# Five-year survival rate and prognostic factors in women with breast cancer treated at a reference hospital in the Brazilian Amazon

Soany de Jesus Valente Cruz[1]*, Andressa Karoline Pinto de Lima Ribeiro[2], Maria da Conceição Nascimento Pinheiro[3], Vânia Cristina Campelo Barroso Carneiro[4], Laura Maria Tomazi Neves[1,2], Saul Rassy Carneiro[1,2]

1 Programa de Pós-Graduação em Ciências do Movimento Humano, Universidade Federal do Pará, Belém, Pará, Brazil, 2 Hospital Universitário João de Barros Barreto, Universidade Federal do Pará, Belém, Pará, Brazil, 3 Programa de Pós-Graduação de Doenças Tropicais, Belém, Brazil, 4 Universidade Federal do Pará, Belém, Brazil

☯ These authors contributed equally to this work.
* cruz.soany@gmail.com

**Data Availability Statement:** All relevant data are within the paper and its Supporting information files.

## Abstract

Breast cancer is the most prevalent malignant neoplasm and the leading cause cancer of death among women globally. In Brazil, survival rates vary according to the region and few studies have been conducted on breast cancer survival in less developed areas, such as the Amazon region. The aim of this study was to analyze the five-year survival rate and prognostic factors in women treated for breast cancer in the city of Belém in northern Brazil. A retrospective hospital-based cohort study was conducted (2007–2013). Sociodemographic, clinical/tumor, and treatment variables were obtained from the records at the Ophir Loyola Hospital. Survival analysis involved the Kaplan-Meier statistical method and Cox regression analysis was performed. The significance level was 5% (p <0.05). A total of 1,430 cases were analyzed. Mean survival time was 51.71 (± 17.22) months, with an estimated overall survival of 79.4%. In the multivariate analysis, referral from the public health care system, advanced clinical stage, lymph node involvement and metastasis were associated with worse prognosis and lower survival rate. Radiotherapy and hormone therapy were associated with increased survival. These findings can contribute to the development of regional strategies for early detection of breast cancer, reducing the incidence and mortality rates and increasing survival time.

## Introduction

Breast cancer (BC) is the most prevalent malignancy and the leading cause of cancer death among women globally, accounting for 14.7% of all cancer-related deaths. The incidence and mortality rate related to BC continue to increase in both developing and developed countries [1, 2]. In Brazil, 15,403 deaths occurred due to BC in 2015 and it is estimated that 59,700 new cases occurred in the period 2018–2019, reaching an estimated risk rate of 56 cases per 100,000 women [3]. In Brazil, according to the Guidelines for the Early Detection of Breast Cancer,

**Funding:** The author(s) received no specific funding for this work.

**Competing interests:** The authors have declared that no competing interests exist.

mammography is routinely recommended for women aged 50 to 69, once every two years. However, due to difficulties in accessing health services, misdiagnosis or delay in diagnosis can occur.

In the northern region of the country, BC is the second most incident form of tumor (19.21/100 thousand) following cervical cancer. The amazon region presents several barriers, from the great distances from the places to the treatment centers to the few specialists in oncology. The state of Pará is the second largest territory in Brazil and has a profound inequality in health services. These difficulties are the biggest problems for patients with breast cancer and make it very important to know the issues related to this population [3].

Several factors are associated with the prognosis of patients with BC, such as clinical stage [4], tumor size [5], and the condition of the axillary nodes [6], along with sociodemographic factors, such as age at diagnosis and education [7, 8]. Although advances in the therapeutic approach are associated with better survival, difficulties in gaining access to health care and a lack of knowledge on the part of the population exert negative impacts, which is why mortality rates for BC tend to be higher in developing countries compared to developed countries [9].

Regarding survival rates, reference countries in Europe and North America achieve five-year survival rates in excess of 80%. In Latin America, the survival rate for BC is approximately 70% [10, 11]. In Brazil, this rate varies among the different regions of the country, with a five-year survival rate of 72.1% in the central western region [12] to 87.7% in in the southern region [13]. However, few studies have analyzed breast cancer survival rates at hospitals in less developed areas of the country, such as the Amazon region.

The aim of the present study was to analyze the five-year survival rate and prognostic factors related to breast cancer in women at an oncological reference hospital in the city of Belém, which is the capital of the state of Pará and is located in the Brazilian Amazon region.

## Materials and methods

A hospital-based retrospective cohort study using medical records was conducted with a sample of women diagnosed with breast cancer between January 2007 and December 2013 at the Ophir Loyola Hospital, which is classified as a high-complexity center in oncology by the National Cancer Institute. All data were fully anonymized before we accessed them and, therefore, the ethics committee waived the requirement for informed consent. This study received approval from the Human Research Ethics Committee of the Ophir Loyola Hospital in the study of Belém, state of Pará (certificate number: 2,682,659).

All female patients with breast cancer as the primary tumor from and living in the state of Pará with the diagnosis confirmed through histopathological or anatomopathological exams and who underwent treatment at the Ophir Loyola Hospital were included. Records with incomplete information were excluded. After searching the Hospital Cancer Registry, the patients were recruited and the necessary information was collected. Patient records were kept anonymous and identification was performed using numerical records.

The following sociodemographic variables were considered: age group (18–29 years, 30–39 years, 40–59, over 60 years), marital status (married, single, divorced, widowed), education (less than eight years, more than eight years), skin color (white, non-white), place of residence (Belém, interior of Pará), smoking (non-smoker, smoker/ex-smoker), and origin (public healthcare system, private healthcare system).

The clinical-tumor characteristics were lymph node involvement (yes, no), clinical stage (I/II, III/IV), histological type (infiltrating ductal carcinoma, carcinoma **in situ**, other histological types), tumor size (smaller than 2 cm, larger than 2 cm), and metastasis (yes, no). The variables related to treatment were time between diagnosis and start of treatment (less than 60 days,

more than 60 days) and the type of therapeutic approach (surgery, radiotherapy, chemotherapy, hormone therapy).

Survival time was calculated as the interval between the date of diagnosis in the hospital record and the date of death or the end of follow-up. The maximum follow-up time was five years. The case of any patient who remained alive after this point was closed.

The data were entered onto spreadsheet of the Excel 2013 program (Microsoft Corporation, CA, USA). SPSS version 22.0 (International Business Machines, NY, USA) was used for data analysis and graphics. The Kaplan-Meier method was used to assess survival probabilities and the comparison of survival functions in relation to the variables was performed using the log-rank test. To estimate the effect of covariates, the Cox model of proportional risks was used, which estimates the proportionality of risks over the entire observation period. The research assumption is that there is an association between socioeconomic and clinical factors with the survival of patients with breast cancer. Variables with a p-value ≤0.20 in the univariate analysis were incorporated into the multivariate model [14]. Chi-square test was used to compare the cases included and excluded from the study. and missing. The significance level was set to 5% (p ≤0.05).

## Results

The search for data returned a total of 2185 cases between 2007 and 2013. 23 records were excluded because of incomplete ascertainment; 78 cases were excluded for uncompleted clinical information (absence date of diagnosis and beginning of treatment) and 611 records were excluded because there were not any information about patient's clinical stage at diagnosis. At the end, 1,430 (65.44%) records were included in the present study. Table 1 shows the comparison between sociodemographic and clinical variables of the included and excluded population.

During the five-year follow-up period of this cohort, there were 294 deaths (20.55%) and the estimated overall survival rate was 79.4%. The mean age at diagnosis was 53.01 (± 13.13) years. Most patients were married (54.3%), had a low education (62.6%) and were non-white (90.5%), presenting lymph node involvement (58.8%), absence of distant metastasis (80.8%) and advanced clinical stage at diagnosis (51.1%). Table 2 display the complete distribution of sociodemographic, clinical-tumor, and treatment variables according to the result of the survival analysis, including the number of cases, number of deaths, survival global, and log rank test.

Regarding the sociodemographic factors, the analysis stratified by education showed that a higher education had a better overall five-year survival rate (84.4%) compared to a low education (73.4%) (p = 0.000). Patients under the age of 30 years had significantly lower survival rate compared to other age groups (p <0.001). Race (p = 0.007) and referral from the public healthcare system (p = 0.001) were also associated with a lower survival rate (Table 2).

Clinical stage was an important factor associated with survival (p <0.001): women diagnosed in early stages had an overall five-year survival rate of 93.4%, whereas those diagnosed in an advanced stage had a survival rate of 66.1%. A significant reduction in survival occurred in cases of lymph node involvement (p <0.001), tumor size larger than 2 cm (p <0.001), and distant metastasis (p <0.001) (Table 2).

Patients who started treatment up to 60 days after diagnosis had a survival rate of 71.8%, whereas those who started treatment after 60 days had a survival rate of 84.4% (p <0.001). The group that started treatment within the 60-day period had worse disease staging at diagnosis (62.5%) compared to those that started treatment after 60 days (41.9%). Radiotherapy (p = 0.012) and hormone therapy (p <0.001) exerted a positive influence on survival, with

**Table 1. Comparison between sociodemographic and clinical variables of the included and excluded population in this study.**

| Variable | Cases included n (%) | Missing category n (%) | p |
|---|---|---|---|
| **Age** | | | 0.371 |
| 18–29 | 30 (2.1) | 21 (2.7) | |
| 30–39 | 181 (12.6) | 87 (11.5) | |
| 40–59 | 775 (54.2) | 392 (51.9) | |
| > 60 | 444 (31) | 255 (31.9) | |
| **Skin color** | | | 0.002 |
| White | 127 (9.5) | 47 (15.6) | |
| Not white | 1207 (90.5) | 255 (84.4) | |
| **Education** | | | 0.090 |
| < 8 years | 785 (62.3) | 361 (58.5) | |
| > 8 years | 469 (37.4) | 246 (41.5) | |
| **Marital status** | | | 0.396 |
| Single | 390 (26.3) | 167 (23.3) | |
| Married | 743 (54.3) | 401(55.9) | |
| Widowed | 174 (12.7) | 103 (14.4) | |
| Divorced | 92 (6.7) | 46 (6.4) | |
| **Place of residence** | | | 0.735 |
| Metropolitan Belém | 978 (68.4) | 551 (67.6) | |
| Interior of Pará | 451 (31.5) | 244 (32.4) | |
| **Smoking** | | | 0.270 |
| Yes | 485 (40.9) | 290 (38.4) | |
| No | 700 (59.1) | 465 (61.6) | |
| **Origin** | | | 0.003 |
| Public healthcare | 657 (49.1) | 402 (57.7) | |
| Private healthcare | 681 (50.0) | 294 (42.3) | |
| **Clinical stage** | | | 0.512 |
| I/II | 699 (48.9) | 38 (45.2) | |
| III/IV | 731 (51.1) | 46 (54.8) | |
| **Histological type** | | | 0.159 |
| Infiltrating ductal carcinoma | 1206 (84.3) | 643 (85.2) | |
| Carcinoma in situ | 16 (1.1) | 15 (1.9) | |
| Others | 208 (14.5) | 97 (12.8) | |
| **Lymph node involvement** | | | 0.953 |
| Yes | 661 (58.8) | 137 (58.5) | |
| No | 464 (41.2) | 97 (41.4) | |
| **Tumor Size** | | | 0.094 |
| < 2 cm | 146 (12.9) | 38 (17.1) | |
| > 2 cm | 985 (87.1) | 184 (82.9) | |
| **Metastasis** | | | 0.347 |
| Yes | 209 (19.2) | 19 (15.7) | |
| No | 878 (80.8) | 103 (84.3) | |
| **Time between diagnosis and treatment** | | | 0.073 |
| < 60 days | 560 (39.2) | 236 (35.1) | |
| > 60 days | 870 (60.8) | 436 (64.9) | |
| **Surgery** | | | 0.251 |
| Yes | 819 (57.3) | 367 (54.6) | |

(*Continued*)

**Table 1.** (Continued)

| Variable | Cases included n (%) | Missing category n (%) | p |
|---|---|---|---|
| No | 611 (42.7) | 305 (45.4) | |
| **Chemotherapy** | | | 0.057 |
| Yes | 1076 (75.5) | 470 (71.3) | |
| No | 354 (25.8) | 189 (28.7) | |
| **Radiotherapy** | | | 0.219 |
| Yes | 808 (56.5) | 325 (53.6) | |
| No | 622 (43.5) | 282 (46.4) | |
| **Hormone Therapy** | | | 0.078 |
| Yes | 412 (28.8) | 219 (32.6) | |
| No | 1018 (71.2) | 453 (67.4) | |

rates of 81.4% and 92.5%, respectively. Chemotherapy had a negative influence (p <0.001), with a survival rate of 76.4% (Table 2).

Based on these results, we performed Cox multivariate analysis to analyze the influence of each independent variable on survival time and found that clinical stage (p = 0.034), referral from the public healthcare system (p = 0.003), lymph node involvement (p = 0.001), metastasis (p <0.001), radiation therapy (p = 0.025), hormone therapy (p = 0.001), and time between diagnosis and treatment (p = 0.002) exerted significant impacts on survival (Table 3). Table 3 displays the final multivariate model of the overall five-year survival of women with breast cancer.

## Discussion

Survival studies are important to determinate the distribution of resources and the identification of the main prognostic factors in a given region and population. However, few studies have analyzed the survival of women with BC in the Amazon region. The five-year survival rate for BC in this study was 79.4%. Other studies conducted in different areas of Brazil report lower rates [12, 15]. Moreover, the survival rate found in this study is higher than that reported for Latin America as a whole (70%), but still lower than that found in certain developed countries (85%) [10, 11].

Having been referred from the public healthcare system was the only socioeconomic variable that significantly influenced survival in this investigation. Similar result was found on a study conducted by the Brazilian Breast Cancer Research Group using different hospital-based records [16]. The worse prognosis of women at public services is related to the diagnosis in advanced stage, with more cases probably detected clinically and fewer detected by screening. Such findings underscore the occurrence of social inequalities and disparities regarding the primary and secondary prevention of BC. Thus, there is a need for structuring and expanding the public network on different levels of health care, with a focus on screening and early detection [17, 18].

Lymph node involvement and the presence of metastases have been associated with a lower survival rate, as reported in previous studies [19, 20]. Regarding the clinical stage, 51.1% of the patients had advanced tumors at the time of diagnosis. Smaller percentages were found in other Brazilian states [21, 22]. A similar pattern was described by Renna Junior and Silva (2018) [23], who state that women treated in the northern region of Brazil are more likely to have advanced tumors at diagnosis compared to those treated in the southern and southeastern regions due to differences in access to healthcare services and the implementation of

**Table 2. Distribution of sociodemographic, clinical and treatment characteristics and survival analysis in women treated for breast cancer.** Belém, Pará, Brazil, 2007–2013.

| Variable | Cases n (%) | Deaths n (%) | Global Survival (%) | Log Rank Test (p) |
|---|---|---|---|---|
| **Age** | | | | <0.001 |
| 18–29 | 30 (2,1) | 15 (50) | 50 | |
| 30–39 | 181 (12.6) | 40 (22.1) | 77.9 | |
| 40–59 | 775 (54.2) | 154 (19.9) | 80.1 | |
| > 60 | 444 (31) | 85 (19.1) | 80.9 | |
| **Skin color** | | | | 0.007 |
| White | 127 (9.5) | 39 (30.7) | 69.3 | |
| Not white | 1207 (90.5) | 236 (19.6) | 80.4 | |
| **Education** | | | | <0.001 |
| < 8 years | 785 (62.3) | 209 (26.6) | 73.4 | |
| > 8 years | 469 (37.4) | 75 (16) | 84% | |
| **Marital status** | | | | 0.548 |
| Single | 390 (26.3) | 79 (21.9) | 78.1 | |
| Married | 743 (54.3) | 155 (20.9) | 79.1 | |
| Widowed | 174 (12.7) | 36 (20.7) | 79.3 | |
| Divorced | 92 (6.7) | 16 (17.4) | 82.6 | |
| **Place of residence** | | | | 0.503 |
| Metropolitan Belém | 978 (68.4) | 197 (20.1) | 79.9 | |
| Interior of Pará | 451 (31.5) | 97 (21.8) | 78.5 | |
| **Smoking** | | | | 0.971 |
| Yes | 485 (40.9) | 99 (20.4) | 79.6 | |
| No | 700 (59.1) | 143 (20.4) | 79.6 | |
| **Origin** | | | | <0.001 |
| Public healthcare | 641(48.5) | 156 (24.3) | 75.7 | |
| Private healthcare | 680 (51.5) | 111 (16.3) | 83.7 | |
| **Clinical stage** | | | | <0.001 |
| I/II | 699 (48.9) | 46 (6.6) | 93.4 | |
| III/IV | 731 (51.1) | 248 (33.9) | 66.1 | |
| **Histological type** | | | | 0.088 |
| Infiltrating ductal carcinoma | 1206 (84.3) | 256 (21.2) | 78.8 | |
| Carcinoma in situ | 18 (1.3) | 0 | 100 | |
| Others | 206 (14.4) | 38 (18.4) | 81.6 | |
| **Lymph node involvement** | | | | <0.001 |
| Yes | 661 (58.8) | 201 (30.4) | 69.6 | |
| No | 464 (41.2) | 30 (6.5) | 93.5 | |
| **Tumor Size** | | | | <0.001 |
| < 2 cm | 146 (12.9) | 0 | 100 | |
| > 2 cm | 985 (87.1) | 231 (23.5) | 76.5 | |
| **Metastasis** | | | | <0.001 |
| Yes | 209 (19.2) | 132 (63.2) | 36.8 | |
| No | 878 (80.8) | 90 (10.3) | 89.4 | |
| **Time between diagnosis and treatment** | | | | <0.001 |
| < 60 days | 560 (39.2) | 158 (28.2) | 71.8 | |
| > 60 days | 870 (60.8) | 136 (15.6) | 84.4 | |
| **Surgery** | | | | 0.115 |
| Yes | 819 (57.3) | 183 (22.3) | 77.7 | |

*(Continued)*

**Table 2.** (Continued)

| Variable | Cases n (%) | Deaths n (%) | Global Survival (%) | Log Rank Test (p) |
|---|---|---|---|---|
| No | 611 (42.7) | 111 (18.2) | 81.8 | |
| **Chemotherapy** | | | | <0.001 |
| Yes | 1076 (75.5) | 254 (23.6) | 76.4 | |
| No | 354 (25.8) | 40 (11.3) | 88.7 | |
| **Radiotherapy** | | | | 0.012 |
| Yes | 808 (56.5) | 150 (18.6) | 81.4 | |
| No | 622 (43.5) | 144 (23.2) | 76.8 | |
| **Hormone Therapy** | | | | <0.001 |
| Yes | 412 (28.8) | 31 (7.5) | 92.5 | |
| No | 1018 (71.2) | 263 (25.8) | 74.2 | |

cancer prevention programs. In general, however, Brazilian women are at greater risk of being diagnosed with advanced breast cancer and at a younger age compared to those in high-income countries [24].

In 2012, a federal law established that cancer patients in the public healthcare system should be treated within an interval not exceeding 60 days (Law N˚ 12.732 2012) [25]. In this study, 60% of the sample started treatment beyond this period. In addition, we found a higher survival rate among those individuals. This result is explained by the clinical stage at diagnosis; 62.5% of the patients who began treatment within the 60-day period were diagnosed in advanced stages compared to 41.9% of those seen after the time. Advanced stage tumors are more resistant to treatment and the therapeutic options are more limited, which increases the risk of death [26]. Results of this nature underscore the need for screening and early diagnostic services in Brazil, where socioeconomic and logistical barriers compromise mammographic screening, resulting in high rates of BC at an advanced stage at diagnosis.

Radiotherapy and hormone therapy were associated with an increase in the survival rate. Adjuvant radiotherapy has been identified as improving the prognosis after surgical treatment and its omission is associated with decreased survival [27, 28]. The use of adjuvant hormone therapy reduces the risk of local recurrence by half and increases survival [29]. A Canadian study concluded that radiotherapy associated with tamoxifen reduces the risk of the axillary recurrence of BC as well as axillary recurrence after quadrantectomy in women with small tumors, negative armpits and positive hormone receptors [30].

This study has some limitations. The retrospective design increases the possibility of missing clinical data in the patient records. As the data were obtained from a secondary source, the analyses were restricted to the information provided in the Hospital Cancer Registry. Moreover, the underreporting of diseases in the state of Pará cannot be ignored, especially in the interior of the state, as a significant number of patients are not registered and die without diagnosis and treatment.

This research has an important role for public health in the Amazon region. The multivariate analysis shows that, referral from the public health system, advanced clinical stage, lymph node involvement and metastasis were associated with worse prognosis and lower survival rate. Radiotherapy and hormone therapy were associated with increased survival. These findings may contribute to the development of regional breast cancer prevention strategies, reducing mortality rates and increasing survival time.

**Table 3. Multivariate analysis of five-year overall survival of women with breast cancer.** Belém, Pará, Brazil, 2007–2013.

| Variable | RR (95% CI) | p |
|---|---|---|
| **Age** | | |
| 18–29 | 1.84 (9.3–3.6) | 0.078 |
| 30–39 | 1.53 (9.3–2.5) | 0.093 |
| 40–59 | 1.20 (9.3–1.7) | 0.321 |
| > 60 | 1 | |
| **Education** | | |
| < 8 years | 1.22 (0.83–1.78) | 0.304 |
| > 8 years | 1 | |
| **Skin color** | | |
| White | 1 | |
| Not white | 0.65 (0.42–1.0) | 0.54 |
| **Origin** | | |
| Public healthcare | 1.66 (1.18–2.34) | 0.003 |
| Private healthcare | 1 | |
| **Clinical stage** | | |
| I/II | 1 | |
| III/IV | 1.66 (1.03–2.67) | 0.034 |
| **Tumor size** | | |
| < 2 cm | 1 | |
| > 2 cm | 921 (000–1934) | 0.919 |
| **Lymph node involvement** | | |
| Yes | 2.35 (1.38–3.98) | 0.001 |
| No | 1 | |
| **Histological Type** | | |
| Infiltrating ductal carcinoma | 0.95 (0.60–1.50) | 0.832 |
| Carcinoma in situ | 0.00 (000–8.87) | 0.968 |
| Others | 1 | |
| **Metastasis** | | |
| Yes | 3.78 (2.64–5.40) | <0.001 |
| No | 1 | |
| **Time between diagnosis and treatment** | | |
| < 60 days | 1 | |
| > 60 days | 0.59 (0.43–0.82) | 0.002 |
| **Hormone Therapy** | | |
| Yes | 0.41 (0.24–0.68) | 0.001 |
| No | 1 | |
| **Radiotherapy** | | |
| Yes | 0.70 (0.51–0.95) | 0.025 |
| No | 1 | |
| **Chemotherapy** | | |
| Yes | 0.75 (0.45–1.23) | 0.259 |
| No | 1 | |
| **Surgery** | | |
| Yes | 0.88 (0.63–1.22) | 0.460 |
| No | 1 | |

RR: Risk Ratio, CI: confidence interval, p <0.05

## Supporting information

**S1 Data.**
(XLSX)

## Acknowledgments

Universidade Federal do Pará—Pró-Reitoria de Pesquisa e Pós-Graduação.

Coordenação de Aperfeiçoamento de Pessoal de Nível Superior (CAPES).

Hospital Universitário João de Barros Barreto–Gerência de Ensino e Pesquisa.

## Author Contributions

**Conceptualization:** Soany de Jesus Valente Cruz, Laura Maria Tomazi Neves, Saul Rassy Carneiro.

**Data curation:** Soany de Jesus Valente Cruz, Andressa Karoline Pinto de Lima Ribeiro.

**Formal analysis:** Soany de Jesus Valente Cruz, Andressa Karoline Pinto de Lima Ribeiro, Maria da Conceição Nascimento Pinheiro, Vânia Cristina Campelo Barroso Carneiro, Saul Rassy Carneiro.

**Investigation:** Maria da Conceição Nascimento Pinheiro.

**Methodology:** Vânia Cristina Campelo Barroso Carneiro, Laura Maria Tomazi Neves, Saul Rassy Carneiro.

**Supervision:** Maria da Conceição Nascimento Pinheiro, Vânia Cristina Campelo Barroso Carneiro, Laura Maria Tomazi Neves.

**Writing – original draft:** Soany de Jesus Valente Cruz, Andressa Karoline Pinto de Lima Ribeiro, Laura Maria Tomazi Neves, Saul Rassy Carneiro.

**Writing – review & editing:** Soany de Jesus Valente Cruz, Andressa Karoline Pinto de Lima Ribeiro, Maria da Conceição Nascimento Pinheiro, Vânia Cristina Campelo Barroso Carneiro, Saul Rassy Carneiro.

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
