## [Decision Letter · Decision Letter 0]

26 Jul 2021

PONE-D-21-12167

Five-year survival rate and prognostic factors in women with breast cancer treated at a reference hospital in the Brazilian Amazon

PLOS ONE

Dear Dr. Valente Cruz,

Thank you for submitting your manuscript to PLOS ONE. After careful consideration, we feel that it has merit but does not fully meet PLOS ONE’s publication criteria as it currently stands. Therefore, we invite you to submit a revised version of the manuscript that addresses the points raised during the review process.

We look forward to receiving your revised manuscript.

Kind regards,

Rubeena Zakar, Ph.D

Academic Editor

PLOS ONE

Additional Editor Comments (if provided):

Reviewers' comments:

Reviewer's Responses to Questions

**Comments to the Author**

1. Is the manuscript technically sound, and do the data support the conclusions?

Reviewer #1: Partly

Reviewer #2: Yes

2. Has the statistical analysis been performed appropriately and rigorously? 

Reviewer #1: Yes

Reviewer #2: Yes

3. Have the authors made all data underlying the findings in their manuscript fully available?

Reviewer #1: Yes

Reviewer #2: Yes

4. Is the manuscript presented in an intelligible fashion and written in standard English?

Reviewer #1: Yes

Reviewer #2: Yes

5. Review Comments to the Author

Reviewer #1: The paper estimates absolute and relative 5-year survival following breast cancer diagnosis in a hospital in the Brazilian Amazon. Only 65% of found cases were included in the analysis. A major concern of the findings is the potential for selection bias differential by patient prognosis.

Major Comments:

1) The authors included 65% (1,430) of the 2,185 cases found in the analysis. This selection needs to be better described.

a) Are these exclusions of records with incomplete information, as per page 3, line 80-81? If so please, provide a breakdown of the numbers excluded because of incomplete mortality ascertainment versus missing covariate information?

b) Please demonstrate that the population excluded is not different from those included in a table 1 comparing characteristics of the two populations (include a missing category) with the appropriate statistical test (chi-square test for categorical variables, Mann-Whitney U test for continuous variables).

Minor comments:

Tables require some double-checking. Comma instead of decimal in the first row of table 1; global survival inconsistently including/not including % in tables 1 and 2; and the decimal places seem to be in the wrong places in the RR column of table 3.

Reviewer #2: The main limitation with this study is that it is hospital-based rather than population-based, nor does there appear to be any population-based data to compare cases in this study with the number of cases in the region. The conclusions of the study therefore are only applicable to women able to access hospital care. Was data on hormone receptor status available? If it is available it would have been an interesting variable to also look at. In model 3 there are 0 cases (seen in Table 2) in some of the cells for tumour size and carcinoma without infiltration . This can make the model unstable. Was any analysis done of correlation between variables? I suspect that some of the variables may be highly correlated and therefore may be able to be excluded from the model.

6. PLOS authors have the option to publish the peer review history of their article (what does this mean?). If published, this will include your full peer review and any attached files.

Reviewer #1: **Yes: **Li C. Cheung

Reviewer #2: No

---

## [Author Response · Author response to Decision Letter 0]

20 Sep 2021

Reviewer: 1

We inform in a new table 1 of the requested information. We compared both populations as requested, and it appeared in table 1. The authors reviewed all the data in the tables as requested.

Reviewer: 2

There were no records on hormonal receptor in the medical records, perhaps because the clinical laboratory of the hospital is not able to do this type of analysis.

About the tumor size in table 2, we noticed that there were no deaths with small tumors, perhaps because it was an early stage of the disease so that it could be more likely to increase survival, but this variable was not important in the adjustment in regression analysis (table 3).

---

## [Decision Letter · Decision Letter 1]

26 Nov 2021

PONE-D-21-12167R1Five-year survival rate and prognostic factors in women with breast cancer treated at a reference hospital in the Brazilian AmazonPLOS ONE

Dear Dr. Jesus Valente Cruz,

Thank you for submitting your manuscript to PLOS ONE. After careful consideration, we feel that it has merit but does not fully meet PLOS ONE’s publication criteria as it currently stands. Therefore, we invite you to submit a revised version of the manuscript that addresses the points raised during the review process.

Please submit your revised manuscript by 7th January 2022. If you will need more time than this to complete your revisions, please reply to this message or contact the journal office at plosone@plos.org. Please include the following items when submitting your revised manuscript:A rebuttal letter that responds to each point raised by the academic editor and reviewer(s). You should upload this letter as a separate file labeled 'Response to Reviewers'.A marked-up copy of your manuscript that highlights changes made to the original version. You should upload this as a separate file labeled 'Revised Manuscript with Track Changes'.An unmarked version of your revised paper without tracked changes. You should upload this as a separate file labeled 'Manuscript'.

We look forward to receiving your revised manuscript.

Kind regards,

Rubeena Zakar, Ph.D

Academic Editor

PLOS ONE

Reviewers' comments:

Reviewer's Responses to Questions

**Comments to the Author**

1. If the authors have adequately addressed your comments raised in a previous round of review and you feel that this manuscript is now acceptable for publication, you may indicate that here to bypass the “Comments to the Author” section, enter your conflict of interest statement in the “Confidential to Editor” section, and submit your "Accept" recommendation.

Reviewer #1: All comments have been addressed

Reviewer #3: (No Response)

2. Is the manuscript technically sound, and do the data support the conclusions?

Reviewer #1: Yes

Reviewer #3: Partly

3. Has the statistical analysis been performed appropriately and rigorously? 

Reviewer #1: Yes

Reviewer #3: I Don't Know

4. Have the authors made all data underlying the findings in their manuscript fully available?

Reviewer #1: Yes

Reviewer #3: Yes

5. Is the manuscript presented in an intelligible fashion and written in standard English?

Reviewer #1: Yes

Reviewer #3: Yes

6. Review Comments to the Author

Reviewer #1: (No Response)

Reviewer #3: Cruz S et al. have reported ' Five-year survival rate and prognostic factors in women with breast cancer treated at a

reference hospital in the Brazilian Amazon'. It is good that the authors try to report an important public health issue in a more neglected area in South America, but there are some weak points in the study that makes it suboptimal:

1. if the idea of authors is to provide an awareness about the status of diagnosis and treatment of BC and yo show their concern about delays they need to explain the currently existing policies and screening programs in the region. Given the specific localization, some information about the culture and the behavior of patients in the region is useful to help to accept the conclusion about the needs for screening and early diagnosis programs

2. There is very poor information about nature of BC tumors in the region in the provided data. There is a general classification of 'infiltrating ductal' versus 'others' with a strange line of 'carcinoma with no infiltration'. What this statement exactly mean? Are the authors talking about carcinoma in situ? Why this needs to be mentioned as an independent category given its very low number (expected). Also it does not fit with the clinical stages provided, CIS of breast is Tis. As soon as breast mass of 1mm, it is already T1 (AJCC, 2018).

3. Infiltrating ductal carcinoma is a big category that covers multiple molecular subtypes that are diagnosed based on IHC and the type of treatment is based on that categorization. Is that mean that in Belem there s no information at all about molecular subtypes or at least the status of HR?

4. Tumor size >2 cm is not really a good indicator of 'T' element in TNM staging, it covers a range of T2 to T4. If such data is not available, then how data on lymph node and metastasis (the combination to define stage of BC) is reliable?

5. It is not clear how the % are calculated: as only one example: The authors report the final sample size (after exclusions) =1430, Table one reports 661 with LN involvement that makes it 46.2%, how the authors end up to 58.8% ? Additionally, how 137 (58.5%) missing data is calculated?

6. Can authors give a short definition about how they categorize the patients to white or non-white and what criteria do they use?

7. PLOS authors have the option to publish the peer review history of their article (what does this mean?). If published, this will include your full peer review and any attached files.

Reviewer #1: No

Reviewer #3: No

---

## [Author Response · Author response to Decision Letter 1]

7 Jan 2022

Reviewer: 1

Reviewer #1: Cruz S et al. have reported ' Five-year survival rate and prognostic factors in women with breast cancer treated at a reference hospital in the Brazilian Amazon'. It is good that the authors try to report an important public health issue in a more neglected area in South America, but there are some weak points in the study that makes it suboptimal:

1. if the idea of authors is to provide an awareness about the status of diagnosis and treatment of BC and yo show their concern about delays they need to explain the currently existing policies and screening programs in the region. Given the specific localization, some information about the culture and the behavior of patients in the region is useful to help to accept the conclusion about the needs for screening and early diagnosis programs.

Our response: We add the requested information.

2. There is very poor information about nature of BC tumors in the region in the provided data. There is a general classification of 'infiltrating ductal' versus 'others' with a strange line of 'carcinoma with no infiltration'. What this statement exactly mean? Are the authors talking about carcinoma in situ? Why this needs to be mentioned as an independent category given its very low number (expected). Also it does not fit with the clinical stages provided, CIS of breast is Tis. As soon as breast mass of 1mm, it is already T1 (AJCC, 2018).

Our response: We used the classification of the Brazilian Society of Clinical Oncology, which classifies the histology of breast cancer between carcinoma in situ and infiltrating carcinoma.

3. Infiltrating ductal carcinoma is a big category that covers multiple molecular subtypes that are diagnosed based on IHC and the type of treatment is based on that categorization. Is that mean that in Belem there s no information at all about molecular subtypes or at least the status of HR?

Our response: Our research used only data from medical records. However, many of these records were old and did not contain this type of information. Furthermore, the hospital does not carry out this type of analysis, being carried out in external laboratories. This analysis is often not attached to the medical record, making it difficult to investigate this topic.

4. Tumor size >2 cm is not really a good indicator of 'T' element in TNM staging, it covers a range of T2 to T4. If such data is not available, then how data on lymph node and metastasis (the combination to define stage of BC) is reliable?

Our response: We have the data available, however we chose to classify the tumor size between smaller and larger than 2cm to establish the tumor at an early or advanced stage in the statistical analysis.

5. It is not clear how the % are calculated: as only one example: The authors report the final sample size (after exclusions) =1430, Table one reports 661 with LN involvement that makes it 46.2%, how the authors end up to 58.8% ? Additionally, how 137 (58.5%) missing data is calculated?

Our response: Some medical records contained information about one variable and not about another. So, the analysis took into account the variable that is contained in the medical record. So, in some cases, the proportions don't match. The missing data corresponds to numbers excluded because of incomplete mortality ascertainment versus missing covariate information

6. Can authors give a short definition about how they categorize the patients to white or non-white and what criteria do they use?

Our response: According to Brazilian law, the identification of race or color must respect the user's self-declaration criteria.

---

## [Decision Letter · Decision Letter 2]

29 Jun 2022

PONE-D-21-12167R2Five-year survival rate and prognostic factors in women with breast cancer treated at a reference hospital in the Brazilian AmazonPLOS ONE

Dear Dr. Valente Cruz,

Thank you for submitting your manuscript to PLOS ONE. After careful consideration, we feel that it has merit but does not fully meet PLOS ONE’s publication criteria as it currently stands. Therefore, we invite you to submit a revised version of the manuscript that addresses the points raised during the review process.

We look forward to receiving your revised manuscript.

Kind regards,

Rubeena Zakar, Ph.D

Section Editor

PLOS ONE

Reviewers' comments:

Reviewer's Responses to Questions

**Comments to the Author**

1. If the authors have adequately addressed your comments raised in a previous round of review and you feel that this manuscript is now acceptable for publication, you may indicate that here to bypass the “Comments to the Author” section, enter your conflict of interest statement in the “Confidential to Editor” section, and submit your "Accept" recommendation.

Reviewer #3: All comments have been addressed

Reviewer #4: All comments have been addressed

Reviewer #5: (No Response)

2. Is the manuscript technically sound, and do the data support the conclusions?

Reviewer #3: Partly

Reviewer #4: Partly

Reviewer #5: Yes

3. Has the statistical analysis been performed appropriately and rigorously? 

Reviewer #3: Yes

Reviewer #4: No

Reviewer #5: No

4. Have the authors made all data underlying the findings in their manuscript fully available?

Reviewer #3: Yes

Reviewer #4: Yes

Reviewer #5: Yes

5. Is the manuscript presented in an intelligible fashion and written in standard English?

Reviewer #3: Yes

Reviewer #4: Yes

Reviewer #5: Yes

6. Review Comments to the Author

Reviewer #3: Thanks for your replies. While your reply to my first review question regarding the current status of BC in the region and a summary of the cultural points that might affect the prognosis of this cancer is 'this data is added', I could not find any addition in the revised version with tracking pointing this question. If you have addressed this question, please be sure if it is done through tracking facility to be found.

Reviewer #4: Thank you very much.

This is an interesting investigation on the breast cancer and the authors have used a survival analysis methodology. The manuscript is generally well structured.

However, it has some shortcomings in regards to some data analyses and narrations.

Below I have provided remarks on the analysis as it is often vague and long-winded.

• The test used is Kaplan-Meier statistical method and Cox regression analysis

However, no information is provided about the assumption tests.

• Why mean survival time? , why not median survival time?

Given these shortcomings the manuscript requires major revisions

Reviewer #5: This article entitled “Five-year survival rate and prognostic factors in women with breast cancer treated at a reference hospital in the Brazilian Amazon” aimed to analyze the five-year survival rate and prognostic factors in Northern Brazilian women from 2007-2013. A total of 1,430 cases were analyzed in this study and in multivariable analysis, referral from public healthcare system, advanced clinical stage, lymph node involvement, and metastasis, were associated with lower survival rates. Overall, the study is important in highlighting this public health issue/disparity in this region of Brazil. While overall the authors did a nice job, there are several areas in the study that could be addressed for clarity.

Abstract/Introduction

1. The authors state in both the abstract and introduction that BC is the leading cause of death among women globally. I would suggest to clarify that to “leading cause of cancer death among women globally”

2. There are font/size discrepancies in the introduction (lines 54-56).

3. Is there a citation for (line 54-56)- early detection of breast cancer in Brazil?

4. What constitutes difficulties in accessing healthcare? Lack of doctors/specialists? Distance to office? Healthcare (public versus private)? Combination of all the above?

Methods

1. Are there any inclusion/exclusion criteria for selection based on clinical factors (other than breast cancer as the primary tumor)?

2. Could be beneficial for clarity to include citation from Hosmer and Lemeshow to describe the univariate variable selection criteria (p<0.2). (Hosmer DW, Lemeshow S, Sturdivant RX. Applied logistic regression. New York: John Wiley & Sons, Incorporated, 2013.)

3. How are deaths defined for the survival analysis? Was it all-cause mortality? Breast cancer related deaths adjudicated by a physician?

4. It may be beneficial to include a section about the variables and how some variables were defined (e.g. why 2 cm tumor size? Is that a proxy to early/late stage clinical stage?)

5. For histological type, why focus on just infiltrating ductal carcinoma and carcinoma in situ? What comprises “others”? Is this due to the detail in the medical records? Any thought about stratifying analysis by histological type for sensitivity analyses?

6. It is not stated in the paper if the proportional hazards assumptions were violated. Also, were there tests for multicollinearity? It seems that several of the tested variables would be strongly correlated (e.g., tumor size & clinical stage; clinical stage & lymph node involvement & metastasis)- all of which made it to the final model.

Results

1. Why the comparison between the missing and included data in Table 1?

2. What source is the global % coming from in Table 2?

3. Argument would benefit from more explanation on the final model (e.g., “lymph node involvement more than doubles the risk of breast cancer in this population (HR= 2.35 [1.38-3.98]; Table 3).”

Discussion

1. There really is no conclusion/summary at the end of the paper

7. PLOS authors have the option to publish the peer review history of their article (what does this mean?). If published, this will include your full peer review and any attached files.

Reviewer #3: No

Reviewer #4: **Yes: **Likawunt Samuel Asfaw

Reviewer #5: No

---

## [Author Response · Author response to Decision Letter 2]

3 Aug 2022

Reviewer #3: Thanks for your replies. While your reply to my first review question regarding the current status of BC in the region and a summary of the cultural points that might affect the prognosis of this cancer is 'this data is added', I could not find any addition in the revised version with tracking pointing this question. If you have addressed this question, please be sure if it is done through tracking facility to be found.

Our response: We add and track the requested information.

Reviewer #4: Thank you very much. This is an interesting investigation on the breast cancer and the authors have used a survival analysis methodology. The manuscript is generally well structured. However, it has some shortcomings in regards to some data analyses and narrations. Below I have provided remarks on the analysis as it is often vague and long-winded.

• The test used is Kaplan-Meier statistical method and Cox regression analysis. However, no information is provided about the assumption tests.

Our response: We add the requested information.

• Why mean survival time? , why not median survival time? Given these shortcomings the manuscript requires major revisions.

Our response: For this case, the estimated survival probability must never have reached 50%, that is, the survival step function does not cross the line y=.5. The Kaplan-Meier estimate, especially since it is a non-parametric method, makes no inference about survival times (i.e., the shape of the survival function) beyond the range of times found in the data. Mean survival time, on the other hand, is a statement about the observed times. It shouldn't be taken to mean the length of time a subject can be expected to survive.

Reviewer #5: This article entitled “Five-year survival rate and prognostic factors in women with breast cancer treated at a reference hospital in the Brazilian Amazon” aimed to analyze the five-year survival rate and prognostic factors in Northern Brazilian women from 2007-2013. A total of 1,430 cases were analyzed in this study and in multivariable analysis, referral from public healthcare system, advanced clinical stage, lymph node involvement, and metastasis, were associated with lower survival rates. Overall, the study is important in highlighting this public health issue/disparity in this region of Brazil. While overall the authors did a nice job, there are several areas in the study that could be addressed for clarity.

Abstract/Introduction

1. The authors state in both the abstract and introduction that BC is the leading cause of death among women globally. I would suggest to clarify that to “leading cause of cancer death among women globally”.

Our response: We correct it. 

2. There are font/size discrepancies in the introduction (lines 54-56).

Our response: We correct it. 

3. Is there a citation for (line 54-56)- early detection of breast cancer in Brazil?

Our response: There is no citation, it is a point of reflection raised by the authors

4. What constitutes difficulties in accessing healthcare? Lack of doctors/specialists? Distance to office? Healthcare (public versus private)? Combination of all the above?

Our response: We add the information required.

Methods

1. Are there any inclusion/exclusion criteria for selection based on clinical factors (other than breast cancer as the primary tumor)?

Our response: The authors chose only this criterion because they aimed to analyze as many variables as possible.

2. Could be beneficial for clarity to include citation from Hosmer and Lemeshow to describe the univariate variable selection criteria (p<0.2). (Hosmer DW, Lemeshow S, Sturdivant RX. Applied logistic regression. New York: John Wiley & Sons, Incorporated, 2013.)

Our response: To build the final regression model and establish the independent variables that influenced survival time, we used the method to include the most important predictors from the p-value (0.2) observed in the univariate analysis. We included the citation.

3. How are deaths defined for the survival analysis? Was it all-cause mortality? Breast cancer related deaths adjudicated by a physician?

Our response: Deaths recorded in the medical records of the breast cancer service were considered. The authors did not distinguish the type of death so as not to incur errors.

4. It may be beneficial to include a section about the variables and how some variables were defined (e.g. why 2 cm tumor size? Is that a proxy to early/late stage clinical stage?)

Our response: The 2cm criterion is established by the staging currently used as a limit. We chose to classify the tumor size between smaller and larger than 2cm to establish the tumor at an early or advanced stage in the statistical analysis.

5. For histological type, why focus on just infiltrating ductal carcinoma and carcinoma in situ? What comprises “others”? Is this due to the detail in the medical records? Any thought about stratifying analysis by histological type for sensitivity analyses?

Our response: We chose this division because it is very precise in terms of the location of the tumor in a single layer of tissue or when the tumor already presents an invasion behavior from other locations (even if they are other layers).

Others means other types of cancer than breast carcinomas, we chose not to stratify this variable too much since they represent 14.5% of the total sample.

It did not occur to the authors to carry out this type of analysis, but we will take this as a suggestion for future research.

6. It is not stated in the paper if the proportional hazards assumptions were violated. Also, were there tests for multicollinearity? It seems that several of the tested variables would be strongly correlated (e.g., tumor size & clinical stage; clinical stage & lymph node involvement & metastasis)- all of which made it to the final model.

Our response: The authors analyzed by the kaplan meier curves, since they are in parallel, the proportional risk is satisfied. We verified by the correlation matrix of the cox model coefficients and observed that there was no correlation above 0.47.

Results

1. Why the comparison between the missing and included data in Table 1?

Our response: The authors wanted to demonstrate the summary of patients who were not involved in the analysis and compare with those who were analyzed.

2. What source is the global % coming from in Table 2?

Our response: Means the survival of each variable stratum

3. Argument would benefit from more explanation on the final model (e.g., “lymph node involvement more than doubles the risk of breast cancer in this population (HR= 2.35 [1.38-3.98]; Table 3).”

Our response: The authors intended to be more objective and allow for a more fluid reading. We will take this suggestion for future manuscripts.

Discussion

1. There really is no conclusion/summary at the end of the paper

Our response: We add the information required.

---

## [Decision Letter · Decision Letter 3]

24 Oct 2022

Five-year survival rate and prognostic factors in women with breast cancer treated at a reference hospital in the Brazilian Amazon

PONE-D-21-12167R3

Dear Dr. Valente Cruz,

We’re pleased to inform you that your manuscript has been judged scientifically suitable for publication and will be formally accepted for publication once it meets all outstanding technical requirements.

Kind regards,

Rubeena Zakar, Ph.D

Section Editor

PLOS ONE

Additional Editor Comments (optional):

Reviewers' comments:

Reviewer's Responses to Questions

**Comments to the Author**

1. If the authors have adequately addressed your comments raised in a previous round of review and you feel that this manuscript is now acceptable for publication, you may indicate that here to bypass the “Comments to the Author” section, enter your conflict of interest statement in the “Confidential to Editor” section, and submit your "Accept" recommendation.

Reviewer #3: All comments have been addressed

Reviewer #4: All comments have been addressed

2. Is the manuscript technically sound, and do the data support the conclusions?

Reviewer #3: Yes

Reviewer #4: Partly

3. Has the statistical analysis been performed appropriately and rigorously? 

Reviewer #3: I Don't Know

Reviewer #4: Yes

4. Have the authors made all data underlying the findings in their manuscript fully available?

Reviewer #3: Yes

Reviewer #4: Yes

5. Is the manuscript presented in an intelligible fashion and written in standard English?

Reviewer #3: Yes

Reviewer #4: Yes

6. Review Comments to the Author

Reviewer #3: Cruz et al analysed the status of BC in a Brazilian amazon region. The authors replied to questions and comments during two level of reviews. Thanks

Reviewer #4: Thank you for revising the manuscript. Most of the comments are addressed, but for clarity I recommend using reading proofing software for the manuscript. Please use median survival time instead of mean survival time is possible.

Good luck!!!

7. PLOS authors have the option to publish the peer review history of their article (what does this mean?). If published, this will include your full peer review and any attached files.

Reviewer #3: No

Reviewer #4: **Yes: **Likawunt Samuel Asfaw

---

## [Editor Report · Acceptance letter]

8 Nov 2022

PONE-D-21-12167R3 

Five-year survival rate and prognostic factors in women with breast cancer treated at a reference hospital in the Brazilian Amazon 

Dear Dr. Cruz:

I'm pleased to inform you that your manuscript has been deemed suitable for publication in PLOS ONE. Congratulations! Your manuscript is now with our production department. 

Kind regards, 

on behalf of

Dr. Rubeena Zakar 

Section Editor

PLOS ONE